# Low-Temperature Hydrothermal Growth of ZnO Nanowires on AZO Substrates for FACsPb(IBr)_3_ Perovskite Solar Cells

**DOI:** 10.3390/nano12122093

**Published:** 2022-06-17

**Authors:** Karthick Sekar, Rana Nakar, Johann Bouclé, Raphaël Doineau, Kevin Nadaud, Bruno Schmaltz, Guylaine Poulin-Vittrant

**Affiliations:** 1GREMAN UMR 7347, Université de Tours, CNRS, INSA Centre Val de Loire, 37071 Tours, CEDEX 2, France; rana.nakar@univ-tours.fr (R.N.); raphael.doineau@univ-tours.fr (R.D.); kevin.nadaud@univ-tours.fr (K.N.); 2Univ. Limoges, XLIM, UMR 7252, 87000 Limoges, France; johann.boucle@unilim.fr; 3CNRS, XLIM, UMR 7252, 87000 Limoges, France; 4PCM2E EA 6299, Université de Tours, Parc de Grandmont, 37200 Tours, France; bruno.schmaltz@univ-tours.fr

**Keywords:** ZnO, nanowires, AZO, hydrothermal growth, perovskite solar cell

## Abstract

Electron and hole transport layers (ETL and HTL) play an essential role in shaping the photovoltaic performance of perovskite solar cells. While compact metal oxide ETL have been largely explored in planar *n-i-p* device architectures, aligned nanowires or nanorods remain highly relevant for efficient charge extraction and directional transport. In this study, we have systematically grown ZnO nanowires (ZnO NWs) over aluminum-doped zinc oxide (AZO) substrates using a low-temperature method, hydrothermal growth (HTG). The main growth parameters were varied, such as hydrothermal precursors concentrations (zinc nitrate hexahydrate, hexamethylenetetramine, polyethylenimine) and growing time, in order to finely control NW properties (length, diameter, density, and void fraction). The results show that ZnO NWs grown on AZO substrates offer highly dense, well-aligned nanowires of high crystallinity compared to conventional substrates such as FTO, while demonstrating efficient FACsPb(IBr)_3_ perovskite device performance, without the requirement of conventional compact hole blocking layers. The device performances are discussed based on NW properties, including void fraction and aspect ratio (NW length over diameter). Finally, AZO/ZnO NW-based devices were fabricated with a recent HTL material based on a carbazole moiety (Cz–Pyr) and compared to the spiro-OMeTAD reference. Our study shows that the Cz–Pyr-based device provides similar performance to that of spiro-OMeTAD while demonstrating a promising stability in ambient conditions and under continuous illumination, as revealed by a preliminary aging test.

## 1. Introduction 

In recent years, several emerging photovoltaic (PV) technologies raised a growing interest in terms of light-weight, cheap process, and high efficiency. Among them, hybrid solar cells using organic–inorganic metal halide perovskites has become one of the most promising topics in material science in the past few years [1,2]. Perovskite photovoltaics studies were triggered by the report on the 9.7% efficient solid-state perovskite solar cell (PSC) in 2012 [2,3]. PSCs composed of organic–inorganic metal halide perovskite have made impressive progress in only a few years, surpassing in efficiencies some well-established thin-film technologies such as CIGS or amorphous silicon [4,5]. In all cases, an electron transport layer (ETL) is used to selectively collect photogenerated electrons from the perovskite absorber layer while blocking holes [2]. The nature of the ETL is also crucial for the perovskite crystallization in *n-i-p* architectures. Considering the optimum specifications of any ETL (high optical transmission for the solar spectrum, suitable electronic configuration with regard to the perovskite for efficient electron extraction, suitable photo- and thermal stability, cheap processing at low temperature < 150 °C, etc.), many organic and inorganic materials have been proposed as a powerful lever to improve the electrical behavior of PSC under working conditions [6,7,8,9]. Among inorganic compounds, titanium dioxide (TiO_2_) is a best-seller ETL material, widely used for efficient PSCs [10,11,12,13]. The appropriate bandgap, high transmittance, and low toxicity of TiO_2_ guarantee a high selectivity for electrons, despite a relatively high processing temperature required to achieve highly crystalline layers. This high-temperature annealing step limits its application for flexible devices and increases the production cost [14]. On the other hand, ZnO was largely considered as a good alternative to TiO_2_ for PSCs since it shows energy levels and physical properties similar to those of TiO_2_, but is more easily processed from solution and at low temperature to achieve structures of different morphologies. Moreover, the intrinsic properties of ZnO thin films and nanostructures can be adjusted by manipulating their morphology, doping, and structural composition [6,15,16,17,18]. 

Recent studies on ZnO-based PSCs have demonstrated high efficiencies and provided many new concepts [11,14]. The increased interest aroused by nanomaterials in ZnO has been largely motivated by its excellent electrical and optoelectronic properties in the bulk, in particular a wide direct forbidden band (3.37 eV) [19], high exciton binding energy (60 meV) [20], and high electronic mobility (200–300 cm^2^/Vs) [21]. A wide variety of ZnO-based structures, such as nanoparticles [11,22], nanowires [23], nanotubes [24], and nanobeads [25], were demonstrated. Nevertheless, ZnO nanoparticles suffer from relatively modest electron transport and high charge recombination due to the presence of numerous grain boundaries and surface defects. In this context, monocrystalline ZnO nanorods provide an easy path for charge transfer due to the absence of grain boundaries, which can delay or inhibit charge recombination [26].

Consequently, in ongoing research efforts for the miniaturization of electronic devices, near-dimensional (1D) ZnO nanowires (NWs) have proven to be potential candidates for ETL due to their unique properties, such as a high electromechanical coupling factor and a better charge injection/extraction at the metal level [27]. Varieties of bottom-up approaches, including pulsed laser ablation [28], flame transport approach [29], vapor-liquid–solid process [30], hydrothermal deposition [31,32,33], were exploited for the synthesis of 1D ZnO NWs. However, most techniques are limited by a high temperature, which cannot be scaled up at a very low cost. The need for an industrially scalable low-temperature method has led to important developments of the hydrothermal growth (HTG) process. HTG is a simplistic, environmentally friendly, cost-effective, straightforward, and low-temperature process (i.e., below 100 °C) in which a 1D monocrystalline material can be produced on various substrates, including plastics and textile fibers [34]. High-density ZnO NWs oriented perpendicularly to the growth substrate have been often reported using the HTG method, for a broad range of applications [35,36,37]. However, only a few papers have explored the complex relationships between NW density, aspect ratio (NW length over diameter), and alignment on their electronic properties, especially for applications in the field of PSC [36,37].

Coming back to the field of perovskite solar cells, the use of ZnO nanostructures as ETL is associated with detrimental degradation mechanisms. Perovskite-based materials are known to be sensitive to water and oxygen in the air but thermally stable under certain conditions [38]. However, according to early reports [24], perovskite materials such as methylammonium lead iodide (CH_3_NH_3_PbI_3_ or MAPI) deposited on ZnO is found to decompose rapidly during heat treatment. Such a phenomenon is not observed between TiO_2_ and MAPI. This behavior is mainly attributed to the different surface properties of ZnO and TiO_2_ nanomaterials. The TiO_2_ surface exhibits a low acidity, while the ZnO surface usually exhibits a basic behavior. Once perovskite is brought into contact with ZnO, there is a deprotonation reaction against the methylammonium cation, which is the basis of instability, and this hypothesis has been proven by theoretical calculations [24].

Another important reason for instability is the presence of chemical residues from manufacturing processes. Materials based on ZnO, in particular nanomaterials used for PSCs, are generally synthesized from chemical methods in solution, which tend to inevitably leave chemical residuals at the final material surface after synthesis. According to Yang et al. and Cheng et al. [24,39], hydroxyl groups and residual acetate in the growth solution exacerbate the degradation of the subsequent deposited perovskite layer. Indeed, the ZnO materials synthesized by the solution process are not completely oxidized, and the surface of the obtained ZnO nanomaterials are covered with oxygenated chemisorbed species such as hydroxide, which could break the ionic interaction between CH_3_NH_3_^+^ and PbI_3_^−^ and destroy the crystal structure of MAPbI_3_ sequentially. This process can be explained by the formula of the following reaction:$${\mathrm{OH}}^{-}+{\;\mathrm{CH}}_{3}{\mathrm{NH}}_{3}I\;\to {\;\mathrm{CH}}_{3}{\mathrm{NH}}_{3}\mathrm{OH}\;+{\;I}^{-}\;$$
$${\mathrm{CH}}_{3}{\mathrm{NH}}_{3}\mathrm{OH}\;\to {\;\mathrm{CH}}_{3}{\mathrm{NH}}_{2}+{\;H}_{2}O\;\left(\mathrm{heating}\right)\;$$

Snaith et al. revealed that the cause of instability of the ZnO/perovskite interface is deprotonation of the methylammonium cation, leading to the formation of zinc hydroxide [40]. Based on this discovery, they replaced the MA cation with formamidinium (FA) of lower acidity, mixed with cesium cations in devices containing ZnO treated at low temperature (120 °C). The stability of the ZnO/perovskite was greatly improved, and an overall power conversion efficiency (PCE) of 21.1% was obtained for the corresponding devices.

In this picture, it is clear that the morphology of the photoanode material plays an important role on the PV performance of ZnO-based PSCs. Achieving well-aligned ZnO nanowires of optimized NW density (number of NWs per unit area) that can efficiently interact with the active layers by preventing charge recombination is a relevant objective to improve device operation. The perovskite infiltration, which should also ensure an optimal light-harvesting ability, is also a crucial parameter. Several groups have demonstrated perovskite solar cells based on such ZnO NW electrodes, which shows that the main growth parameters, such as hydrothermal precursor ratios and growing time, and growth solution concentration, significantly influence NWs’ properties (i.e., NW length, diameter, density, and morphology) [41,42,43,44,45,46] and the device PV performances [17,23,47].

In this paper, ZnO nanowires grown on aluminum-doped ZnO (AZO) substrates by HTG method are carefully investigated with regard to the specific application as ETL in *n-i-p* perovskite solar cells. While the application drives specific features for efficient charge extraction, we first demonstrate that the use of AZO as growing substrates does not require any additional compact or seed layer for their growth, leading to a simplified fabrication process and device architecture. We then systematically investigate the influence of the main ZnO NW growth parameters, such as hydrothermal precursor concentration and growing time, on the NW properties, before discussing the impact on the performance of perovskite solar cells fully prepared under ambient conditions. Finally, optimized ZnO NW arrays deposited on AZO are combined with a novel carbazole hole transport material derivative (Cz–Pyr as HTL), demonstrating promising performance compared to the reference spiro-OMeTAD. 

## 2. Experimental Details

### 2.1. Materials

Commercial AZO-coated glass substrates were purchased from MSE Supplies (Tucson, AZ, USA). Zinc nitrate hexahydrate Zn(NO_3_)_2_.6H_2_O (98%), hexamethylenetetramine (HMTA) (CH_2_)_6_N_4_ (>99.5%) purchased from Sigma-Aldrich (St. Louis, MI, USA), and polyethylenimine branched (PEI) (average Mw ~25,000) from Sigma-Aldrich, also used as received. These materials were used for the preparation of ZnO NWs. The deposition of ZnO seed layer was done with physical vapor deposition (PVD) equipment (Plassys MP 550 S, Marolles-en-Hurepoix, France). A tubular furnace (Thermolyne 79300, Dubuque, IA, USA) was used for the ZnO seed layer annealing treatment. A stainless-steel autoclave was used to operate the synthesis of ZnO NWs. For perovskite solar cells: Formamidinium iodide (FAI) (≥90%, GreatCell Solar, Queanbeyan, Australia), cesium iodide (CsI) (99.9%, trace metals basis, Sigma Aldrich, St. Louis, MO, USA), lead iodide (PbI_2_) (99%, Sigma Aldrich), lead bromide (PbBr_2_) (≥90%, Sigma Aldrich), tin oxide nanoparticle colloidal solution (SnO_2_) (15% in H_2_O colloidal dispersion, Alfa Aesar (Haverhill, MA, USA)), spiro-OMeTAD (99% HPLC, Sigma Aldrich), 4-tert-butylpyridine (96%, Sigma Aldrich), chlorobenzene (CB, anhydrous, 99.8%, Sigma Aldrich), N-N dimethylformamide (DMF, anhydrous, 99.8%, Sigma Aldrich), dimethyl sulfoxide (DMSO anhydrous, ≥99.9%, Sigma Aldrich), acetonitrile (anhydrous, 99.8%, Sigma Aldrich), diethyl ether (DE, ≥99.5%, GC, Sigma Aldrich).

### 2.2. ZnO Nanowire Growth

ZnO nanowires were synthesized on all kinds of substrates via the hydrothermal growth route [48]. All the substrates, except AZO, were precoated by a ZnO seed layer (50 nm) prepared by radio-frequency sputtering during 16 min at 65 Watts and under a pressure of 5 mTorr in an argon atmosphere. Then, a thermal treatment was performed at 400 °C for 1 h in air through a horizontal quartz tubular furnace. The hydrothermal synthesis of ZnO NWs was carried out in the stainless-steel autoclave. Two clear and transparent fresh stock solutions of Zn(NO_3_)_2_ and HMTA were separately prepared in distilled water (DI), as well as a PEI solution. The concentrations of each solution were adjusted for each specific case, as described in the next sections. To hydrothermally grow the ZnO NWs, the autoclave was filled with the solutions following this order: zinc nitrate hexahydrate and HMTA, and PEI before stirring. The substrates were tilted against the walls to prevent the precipitation of homogenous nucleated ZnO on the seed layer surface. After hermetically sealing the autoclave in order to avoid evaporation, the temperature was maintained at 90 °C during the growth process. Finally, the samples were rinsed with distilled water and dried under airflow. Then, thermal treatment was performed on the ZnO NW coated samples at 450 °C for 30 min in air through a horizontal quartz tubular furnace.

### 2.3. Perovskite Solar Cells Fabrication

The FA_0.85_Cs_0.15_Pb(I_0.85_Br_0.15_)_3_ precursor solution was prepared by mixing FAI (0.1463 g), PbI_2_ (0.392 g), PbBr_2_ (0.0554 g), CsI (0.039 g), and DMSO (78 μL) in the DMF (600 μL), which was spread on the substrate and spun at 4000 rpm for 30 s [49]. While spinning, diethyl ether was dripped to induce adduct intermediate by eliminating DMF. The as-coated perovskite film was annealed at 100 °C for 20 min. The spiro-OMeTAD solution was prepared by dissolving 36.6 mg of spiro-OMeTAD in 0.500 mL chlorobenzene. Classical additives were added to the spiro-OMeTAD solution, namely 8.75 μL LiTFSI (520 mg in 1 mL acetonitrile) and 14.4 μL *t*BP. Then, the spiro-OMeTAD layer was formed by spin coating at 3000 rpm for 30 s on top of the perovskite active layer. For comparison, an alternative hole transport layer made of 3,6-Bis[N,N′-di(4-methoxyphenyl)amino]-9-(1-pyrenyl)carbazole (Cz–Pyr) was prepared by dissolving 49 mg Cz–Pyr in 0.500 mL chlorobenzene, where 8.75 μL LiTFSI and 14.4 μL *t*BP were added. Cz–Pyr HTL was recently demonstrated as a relevant alternative to spiro-OMeTAD, leading to similar photovoltaic performance [50]. Finally, an Au electrode (~100 nm) was deposited by using thermal evaporation under secondary vacuum through a shadow mask, defining the active area of the cell (>0.22 cm²). 

### 2.4. Characterization Techniques

The surface morphology of the ZnO NWs was measured using a field emission scanning electron microscope (SEM, Hitachi S-4800). Lengths were measured using cross-sectional SEM images, while diameters were estimated using top-view images. The average NW dimensions (length and diameter), the NWs’ density, and the void fraction were calculated from simple SEM image analysis through the ratio of the surface occupied by ZnO to the total surface of the substrate. Average values were obtained from measurements made on a large number of ZnO nanowires on a single AZO substrate. The current density–voltage (J–V) characteristics of photovoltaic devices were measured in ambient air (25–35 °C) on unencapsulated devices using a solar simulator (1600 W NEWPORT) equipped with an AM 1.5 G filter and a source meter (Keithley 2400). The active area of the devices was defined by a masked aperture of approximately 0.22 cm^2^, and spectral mismatch correction was systematically applied, and all the characterizations were carried out in ambient air without encapsulation (moisture level of 55–60%). Several devices (>3) presenting comparable features have been characterized in each case, both in forward and reverse scans (simply FS and RS, voltage scan in the order of 50 mV/s), in order to extract the dispersion of performance. The shunt and series resistances (simply—R_Sh_ and R_S_) were extracted from the simple estimation of the J–V curve slopes in the open- and short-circuit conditions, assuming R_Sh_ ≫ R_S_. The external quantum efficiency (EQE or IPCE) was measured under steady-state conditions using a continuous monochromated xenon lamp and a calibrated pico-amperemeter (Keithley 485). A reference silicon photodiode of known spectral sensitivity was used for EQE calibration.

## 3. Results and Discussion

Hydrothermal growth was used to synthesize ZnO NWs electrodes subsequently integrated into PSCs (see experimental section). The growth parameters (growing time, precursor’s concentration, and growth temperature) were varied to identify the optimal features in the context of the photovoltaic application. We remind below the specific mechanisms involved during the ZnO NW formation on a ZnO-based growing substrate (either on AZO or ZnO seed layer), which can be summarized by the following equations [51,52]:C_6_H_12_N_4_ + 6H_2_O ⟶ 4NH_3_ + 6HCHO(1)
NH_3_ + H_2_O ⟶ NH_4_^+^ + OH^−^(2)
Zn^2+^ + 4OH^−^ ⟶ Zn(OH)_4_^2−^(3)
Zn(OH)_4_^2−^ ⟶ ZnO + H_2_O + 2OH^−^(4)

During hydrothermal growth, ZnO NWs develop following a series of reactions (Equations (1)–(4)). HMTA is initially hydrolyzed with heat, forming formaldehyde (HCHO) and NH_3_ (Equation (1)). HCHO does not participate directly in the growth of NWs but reacts indirectly, as will be explained in the following parts. Then, NH_3_ protonates, producing OH^−^ ions (Equation (2)). The Zn^2+^ ions formed following the solubilization of Zn(NO_3_) react with the OH^−^ ions to form Zn(OH)_4_^2−^ (Equation (3)), which dehydrates, leading to the direct crystallization of ZnO (Equation (4)). Certain milky white precipitation from homogeneous nucleation is present in the reaction medium. This will be explained in the following parts. Finally, the samples are rinsed with distilled water and dried under airflow.

In the following sections, we first discuss the influence of the growing substrate before focusing on ZnO NW forest grown on AZO to systematically point out the influence of the growth parameters on the NW properties and associated photovoltaic performance of devices.

### 3.1. Photovoltaic Performance as a Function of Growing Substrate and ETL

We all know that the solar cell performance does not solely depend on any layer (i.e., TCO, ETL, Absorber, HTL, or electrode) because in order to get a high power conversion efficiency, each layer arrangement in the device is crucial without any defects. Usually, most state-of-the-art PSCs are based on SnO_2_-ETL on which the perovskite can be grown [53]. The ETL plays a role in the extraction of the negative charge carriers to the cathode and also strongly impacts the perovskite active layer morphology as well as the quality of the ETL/perovskite interface. The ETL also brings a specific processing step that has to be taken into account for large-scale developments. In the case of ZnO NWs, the hydrothermal growth required a starting ZnO seed layer, which can be formed by various techniques on different substrates, including conventional FTO-coated glass [41,54,55,56]. However, zinc–oxide-based transparent electrodes such as AZO have been suggested to potentially act as efficient seed layers [48,57,58]. For these reasons, the performance of devices based on NW grown on different substrates have been compared: FTO/compact ZnO (referred as ZnOc) or bare AZO (both without ETL). We also include comparison with planar devices based on various ETL without NW: FTO/SnO_2_, AZO/ZnOc, or AZO/SnO_2_. NW-based devices were processed from NWs grown using a set of reference experimental parameters (precursor concentration ratio of Zn(NO_3_)_2_ to HMTA and PEI fixed to [Zn^2+^]/[HMTA] = 1 and [Zn^2+^]/[PEI] = 300, growing time of 6 h and 15 h), which will be subsequently tuned in the next sections of this article on the best performing substrate. After NW-based ETL deposition, the mixed cation halide perovskite active layer FA_0.85_Cs_0.15_Pb(I_0.85_Br_0.15_)_3_ was deposited from the solution before the spiro-OMeTAD HTL and gold top electrode (see full procedure in the experimental section). Avoiding methylammonium was necessary in order to prevent the rapid degradation of the ZnO/perovskite interface, as discussed in the introduction section. The photovoltaic parameters, extracted from current–voltage characteristics recorded under simulated standard illumination (Appendix A), are given as Appendix A. First, we observe that the classical planar architecture based on the FTO/SnO_2_ ETL is associated with the highest performance (up to nearly 14% efficiency, considering that no passivation strategies are used), as reported in previous work [49,59]. Besides, a clear S-shape in the J–V curve (reverse scan) is observed for AZO/SnO_2_ device (Appendix A), which is not the case of any devices based on ZnO ETLs (neither compact layer nor NWs). This observation is consistent with previous reports made in the literature, emphasizing that the AZO/ZnO interface can easily generate an ohmic contact due to the inherent nature of the materials in both cases (zinc oxide), which facilitates charge extraction [60]. The electrical characteristics of the AZO/SnO_2_-based device suggest a strong limitation for charge extraction at this interface, which is not the case of the reference devices fabricated on fluorinated tin oxide (FTO) substrates. In Appendix A, we clearly observe that the presence of the ZnO NW is required to achieve reasonable current densities up to ~20 mA cm^−2^, which remain reduced with regard to the FTO/SnO_2_ reference. This observation can be associated with a better charge extraction efficiency generally attributed to NWs compared to compact layers, which agrees with previous reports [61,62]. However, a much lower open-circuit voltage ($V_{oc}$) has been observed for ZnO-based devices compared with SnO_2_ ETL devices (Appendix A). This could be due to the inherent recombination losses associated with the ZnO/perovskite interface. Overall, on AZO substrate better device performances are displayed with ZnO NWs electrode rather than compact layers made of ZnO or SnO_2_ (see Appendix A), especially due to a favorable electrical contact with the electrode and to the positive influence of the NW aspect ratio on charge extraction (clearly evidenced through a much lower series resistance for the AZO/ZnO-NW device with regard to the other AZO-based devices). These results clearly demonstrate the relevance of using bare AZO electrode to grow the NW, which, in addition, strongly simplify the overall fabrication protocol of the devices. This growing substrate has been chosen for the following studies. However, a low shunt resistance and fill factor suggest room for improvements as a function of NW properties. In the following sections, we will explore the influence of the precursor concentration and growing time of the NW related to the properties. These devices, made of ZnO NW grown on AZO substrates, will be discussed in terms of achieved PV performances. 

### 3.2. Influence of ZnO NWs’ Growing Time 

Figure 1 shows the SEM images of samples prepared with different growing times from 2 h to 15 h (i.e., 2 h, 4 h, 6 h, and 15 h) with fixed precursor concentration ratio of Zn(NO_3_)_2_ to HMTA and PEI (([Zn^2+^]/[HMTA] = 1), [Zn^2+^]/[PEI] = 300). The average NWs’ dimensions (length and diameter), the NWs’ density, and the void fraction are summarized in Table 1. 

Considering the length of the NWs, the growing time parameter has an important impact, and we can see the quite different morphological behavior, especially in the NWs’ length, diameter, and void fraction values (Table 1). Indeed, the length increases from 0.1 to 1.3 μm by increasing the growing time from 2 h to 15 h. In addition, the growing time also has an influence on the NW density. For short growing times (i.e., 2 h and 4 h), very dense arrays are obtained, and the NWs are largely interconnected in the plane of the substrate (Figure 1a,b). As a result, no significant void fraction could be measured at such a short growing time. For a longer growing time (6 h and 15 h), a larger axial growth led to an increase in the void fraction from 28% to 58%, respectively. Consequently, a decrease of the NW density as a function of the growing time is clearly evidenced, while the average diameter of the NWs also increased from 127 nm after 6 h to 165 nm after 15 h, but at a much slower rate than the length, which is attributed to the presence of PEI hindering only the lateral growth of the ZnO NWs [63].

Considering the global morphology of the obtained ZnO NW arrays, we focused on samples grown for 6 h and 15 h to fabricate *n-i-p* perovskite solar cells of structure glass/AZO/ZnO NW/perovskite/spiro-OMeTAD/Au. Arrays grown at shorter times (i.e., 2 h and 4 h) did not enable a suitable perovskite deposition, making the investigations unworthy (not shown here). The UV–visible absorption and transmittance spectra of bare AZO, AZO/ZnO NW (6 h and 15 h), before and after perovskite deposition are presented as Appendix A. A slight decrease in transmittance is observed after NW growth compared to bare AZO substrate, with transmission over 85% in the visible spectrum in all cases. After the perovskite deposition, a broad and intense absorption extending up to the near-infrared is observed, which is typical of the perovskite material used in this work. We note that a smaller absorption seems to be observed on the longer NW grown for 15h, which seems to indicate a difficulty for perovskite deposition in this case. 

Figure 2 shows the forward and reverse scan (FS and RS, respectively) current–voltage characteristics under simulated solar illumination (AM1.5G, 100 mW cm^−2^) of devices based on ZnO NW arrays as a function of growing time. We can clearly understand that it is challenging to get the consistent solar cell performance, even with the same batch of samples (see Appendix A), due to several factors such as interface defects, layer arrangement, perovskite infiltration, voids/pinholes, etc. Only champion cells are presented here, but several devices were characterized in each case (see Appendix A), showing similar trends, and the corresponding photovoltaic parameters are presented in Table 2.

Both cells are showing a clear photovoltaic effect with reduced hysteresis. Clearly, a longer growing time (15 h) is detrimental to device efficiency, mainly through reduced J_sc_ and V_oc_ values. Such observation seems consistent with the optical absorption measurements, as a reduced light-harvesting efficiency for the device based on the long NW was evidenced. It can suggest a more difficult crystallization for perovskite, which in turn leads to poor photovoltaic parameters (poor photocurrent). The SEM cross-section image of a device based on the short NW grown for 6h supports this assumption (see Appendix A), as a well-defined sandwiched structure is observed, with the presence of compact perovskite grains on the ZnO NWs array. This performance variation between 6 h and 15 h devices might be due to the huge gap between nanowires without a proper alignment (Figure 1c,d), which possibly creates a direct contact between the perovskite and the AZO layers. And also, the 15 h-grown NWs length is higher (1.3 µm) than in the 6 h device, so the longest NWs can possibly penetrate the perovskite layer and make a direct connection with HTL (spiro-OMeTAD), leading to short-circuit current and recombination issues in the devices. Moreover, while having a higher NW diameter, it possibly does not offer a sufficient gap for the perovskite infiltration that restricts the charge extraction (electron), which causes the PV performance reduction. A much larger series resistance is also evidenced for longer NW, which can be associated with poorer charge transport properties, as reported in the literature for similar ZnO NWs [17,23,47]. In this present case, these preliminary observations indicate that a growing time of 6 h is in any case suitable as a starting point for further investigation of the different growth parameters governing the morphology and properties of the ZnO NWs.

### 3.3. Optimization of the Hydrothermal Process

In the following sections, we systematically investigate the effect of the main chemical parameters governing the growth of ZnO NWs, including the PEI, HMTA contents, and the growth solution concentration (related to Zn^2+^ quantity). Their influence on the length, density, and void fraction of the ZnO NWs will be discussed. The main objective is to optimize the precursor concentrations in order to achieve a suitable electrode morphology that favors the perovskite solar cell efficiency. In the following sections, the impact of the chemical parameters on the NW properties will be discussed. Then, the relation between the ZnO NW properties and the device performance will be presented. 

#### 3.3.1. Effect of PEI Content

The effect of the PEI content on the growth of ZnO NWs is already discussed in the literature, and the studies were devoted to NWs that are grown on the ZnO seed layer [46,64,65,66] or on the FTO layer [26]. They show that PEI addition significantly enhances the length of the NWs, while at the same time, it diminishes their aspect ratio [64,67,68]. In this work, we focus on the effect of the PEI content on the growth of NWs using AZO substrates as seed layers. Such a strategy aims to reduce the device fabrication complexity as no seed layer is required in this case. The ZnO NWs were synthesized at 90 °C for 6 h (temperature, growing time, and Zn(NO_3_)_2_ concentration 50 mM has been fixed) using various PEI contents ranging from 0 to 13 mM. The morphology characteristics deduced from SEM images (see Appendix A) are presented in Table 3. The PEI to HMTA ratio is also given as it obviously evolves as a function of PEI content. 

For the reference sample without PEI, short ZnO NW with an average length of 0.3 μm is obtained. As the PEI content increases up to 11 mM, the length of the NWs increases to reach a maximum length of 0.5 μm. Adding over 11 mM (i.e., 13 mM), NW length decreases to 0.2 μm, reaching a shorter length than without using PEI. Additionally, the average void fraction of the NWs tends to increase with increasing PEI content (from 5 to 11 mM) with a maximum of 45%. Still, increasing PEI concentration to 13 mM implies a decrease of the parameters, and the void fraction decreases to 28%, and this trend is similar to that of previously published reports [64,68]. Also, the results are consistent with the fact that PEI mainly hinders the lateral growth of the wires and favors their axial growth in solution while maintaining a relatively high NW density. Therefore, considering the requirements suitable for perovskite photovoltaics, an appropriate void fraction is expected to favor the perovskite infiltration for an intimate electronic contact with the nanostructured ZnO [69]. In our study, the PEI concentration leading to the maximal void fraction is at 11 mM. In the next section, we will keep this parameter fixed to explore the influence of HMTA and zinc salt concentration on the NW morphology. 

#### 3.3.2. Effect of HMTA Content

Fixing the PEI concentration fixed at 11 mM, the Zn(NO_3_)_2_ to HMTA ratio has been tuned. SEM images (see Appendix A) are used to extract the main morphologic features, which are summarized in Table 4 as a function of Zn(NO_3_)_2_ to HMTA ratio (from 1 to 2). This ratio was adjusted through the HMTA concentration from 50 to 25 mM. As previously mentioned, the concentration ratio [Zn(NO_3_)_2_]/[PEI] is fixed to 4.5.

The length of the obtained ZnO NWs decreases from about 0.5 to 0.3 μm as the [Zn(NO_3_)_2_]/[HMTA] ratio increases from 1 to 2. The evolution of the diameter of the wires gives additional important insight into the role of HMTA in solution: the NW diameter decreases from ~125 to 101 nm as the HMTA content decreases. Using less HMTA, therefore, decreases both the NW diameter and length. Let us remember that Zn^2+^ ions are formed in solution following the solubilization of Zn(NO_3_). Then these ions react with the hydroxide OH^−^ ions (resulting from NH_3_ protonation) to form Zn(OH)_4_^2−^ (Equation (3)). This latter species dehydrates, leading to the direct crystallization of ZnO, whereas OH^−^ ions and NH_3_ play the role of ligands to form Zn(II) hydroxide and amine complexes. HMTA is initially hydrolyzed with heat, forming formaldehyde (HCHO) and NH_3._ In precursor solution based on usual conditions, the produced HCHO does not take part in the growth, while in the presence of PEI, the HCHO molecules can react with –NH_2_ groups, releasing Zn^2+^ ions chelated by PEI. –N=CH- or –N=CH_2_ bonds are produced by the reaction between PEI and HCHO. The possible reaction between PEI and HCHO can be described in Figure 1 according to the typical Mannich reaction between the –CH=O group and the –NH_2_ or –NH- group [70].

This suggests that PEI addition has made Zn^2+^ the limiting species for the growth instead of NH_3_. Meanwhile, increasing the HMTA concentration (which increases the HCHO concentration) leads to a faster Mannich reaction. This faster reaction brings more Zn^2+^ ions into the solution, increasing the growth rate of ZnO NW. Hence, when the [Zn(NO_3_)_2_]/[HMTA] ratio is small, the limiting reactants for the axial growth of ZnO NWs are OH^−^ and Zn^2+^ ions, resulting in a fairly small axial and radial growth by inhibiting the development of their sidewalls [71,72]. 

This may explain our results, showing that, for [Zn(NO_3_)_2_]/[HMTA] ratio equal to 1.33, ZnO NWs present a compromise between a high length and a small diameter, corresponding to a high aspect ratio (see Table 4). This morphology also results in a relevant void fraction compared to a [Zn(NO_3_)_2_]/[HMTA] ratio of 1, with a constant density. Therefore using non-equimolar growth conditions with an excess of Zn(NO_3_)_2_ is required to reach an optimum morphology, especially with smooth surface without pinholes or voids. These conditions can be summarized as: [Zn(NO_3_)_2_] = 50 mM, [PEI] = 11 mM, and [HMTA] = 37.5 mM, for growing time of 6 h at 90 °C, and 0.5 μm long ZnO NWs have been obtained in such conditions (see Appendix A). Such a length might appear too large for the application, considering the electrical shunt resistance that was pointed out in the previous section. Therefore, a study of the role of the zinc solution was conducted, and the aim was to try to control the NWs’ length while keeping a void fraction over 35%. 

#### 3.3.3. Effect of Zn(NO_3_)_2_ Concentration

ZnO NWs were fabricated from various zinc salt concentrations [Zn^2+^] ranging from 20 to 50 mM, using fixed concentration ratios between Zn(NO_3_)_2_ and HMTA ([Zn^2+^]/[HMTA] = 1.33) and between Zn(NO_3_)_2_ and PEI ([Zn^2+^]/[PEI] = 4.5). SEM images were obtained (see Appendix A) and analyzed to extract the main morphologic features of the samples (Table 5). 

As deduced from SEM pictures, the length of the NWs decreases with decreasing zinc nitrate concentration. For a concentration of 20 mM, the length is 0.25 μm (growing time 6 h) and increases up to 0.5 μm for a concentration of 50 mM. We emphasize that well-aligned NWs are obtained in this latter case (without any pinholes, see Appendix A), leaving a comparable porosity of 36% (void fraction) compared to 20 mM and 35 mM of Zn(NO_3_)_2_ contents. Indeed, when the reagent concentration is increased, the transport rate of active ions increases too. It results in a relatively high nanowire growth rate in the axial direction, as seen with 50 mM of Zn(NO_3_)_2_ [64]. This study confirms that the length of the NWs increases with the concentration of the zinc salt. 

Considering our initial assumption about the adverse influence of NWs’ length on perovskite solar cell performance (refer to data from Section 0: influence of growing time), we now focus on devices based on NWs grown using different zinc salt concentrations. This simple approach aims to decipher the relation between PV performance and the morphological characteristics of the integrated ZnO NWs. Figure 3 shows the current density–voltage (J–V) characteristics under the illumination of perovskite solar cells based on ZnO NWs grown using various zinc salt concentrations, and the corresponding PV parameters are summarized in Table 6. 

Clearly, by decreasing the zinc salt concentration down to 20 mM, better device performance is observed. A maximum power conversion efficiency of 2.7% was achieved for the PV cell corresponding to this zinc salt concentration (hence corresponding to the shorter NWs), mainly due to a combined high short-circuit current and open-circuit voltage. In general, the NWs’ size and length should be controlled; otherwise, the perovskite cannot be well-deposited into the NW gaps or surface, which possibly creates unsmooth morphology, and increases chance of defects such as voids or pinholes in the layer [73]. This exploration of the main growth parameters enables us to point out suitable conditions for the synthesis of relatively short NW presenting a suitable void fraction, which is found to be critical to ensure suitable perovskite layer infiltration. This better infiltration may reduce the occurrence of direct shortcuts with the HTM. The main morphologic features of the ZnO NWs grown using all sets of parameters are summarized in Appendix A. After careful adjustment of the main parameters of the ZnO NWs’ growth, the final set ([Zn(NO_3_)_2_] = 20 mM, ([Zn^2+^]/[PEI] = 4.5; [Zn^2+^]/[HMTA] = 1.33, growing time of 6 h) is leading to a PV cell performance that is lower than the initial set ([Zn(NO_3_)_2_] = 50 mM, [Zn^2+^]/[HMTA] = 1 and [Zn^2+^]/[PEI] = 300, for 6 h growing time). The following Table 7 summarizes the corresponding PV cell characteristics.

It is worth mentioning here that while varying NW growing time, precursor concentrations (zinc nitrate hexahydrate, HMTA, and PEI), the obtained ZnO NWs’ properties, especially NW length and diameter values, were associated and are consistent with the range of previously published reports (see Appendix A) [23,41,42,43,44,45,46,63,64,74]. The ZnO NWs’ growing time should be limited to 6 h (0.4 μm long ZnO NWs) because longer NWs seem detrimental to the PV cell performance (Table 2). With our growth parameters, a PEI content of 11 mM seemed to be the optimal point to maximize the void content (here 45%) while keeping a limited length of the NWs (0.5 μm) (Table 3). In order to keep a good surface morphology with an appropriate void content, but also to decrease the ZnO NWs’ length, the HMTA concentration effect was also studied. Apart from the NW length (0.5 μm), a smoother surface behavior was obtained for 37.5 mM than for other concentrations with a void fraction value of 36% (Table 4 and Appendix A). Finally, the Zn(NO_3_)_2_ zinc salt content was studied in order to fine-tune the NWs’ length while trying to maintain a suitable void fraction (Table 5). This study showed that the void fraction decreases additionally to the length: for example, 20 mM zinc salt concentration led to 0.25 μm long NWs and a 43% void fraction compared to 35 mM, and this behavior is not same for the higher concentration (i.e., 50 mM). Therefore, the PV cells resulting from this last study seem to show that short ZnO NWs are prioritary to low dense NW arrays in order to maximize the PV cell efficiency.

In order to reinforce this conclusion and to get a deeper overview of the design choices, we decided to conduct a final study, by comparing performance of devices made with both initial and final sets of growth parameters and with two different HTLs. Consequently, in the following section we present a preliminary study over the performance of ZnO-NW arrays associated with an alternative HTL based on a carbazole moiety (Cz–Pyr).

### 3.4. Comparison between Spiro-OMeTAD and Cz–Pyr as HTL

As previously described, we succeeded to control the morphology parameters of ZnO NWs grown on AZO substrate. In order to discuss the relation between the morphology and the PV efficiencies, devices with ZnO NWs have been tested. The devices have been fabricated using two different HTG parameters: on one side the initial chemical process, and on the other side, the optimum recipe we have selected from the previous studies. Moreover, the devices have been fabricated using two different HTLs. Carbazole-based derivatives as HTL in perovskite solar cells have attracted attention because of their photochemical properties and their promising characteristics such as high mobilities, good charge transport, and high efficiencies [75,76,77,78]. In this context, reported Cz–Pyr [50] has been compared with the reference spiro-OMeTAD as HTL. In short, in this subsection, optimized NW growth conditions (i.e., for 20mM, [Zn^2+^]/[HMTA] = 1.33 and [Zn^2+^]/[PEI] = 4.5, for 6 h growing time) and initial NW growth conditions (i.e., for 50 mM [Zn^2+^]/[HMTA] = 1 and [Zn^2+^]/[PEI] = 300, for 6 h growing time) have been used to fabricate perovskite solar cells based on either spiro-OMeTAD or Cz–Pyr. Especially, two different Zn(NO_3_)_2_ concentrations, either [Zn(NO_3_)_2_] = 20 mM (NW referred in this case of ZnO—0.25 µm) or [Zn(NO_3_)_2_] = 50 mM (NW referred in this case as ZnO—0.40 µm), have been tested. We especially compare two NW lengths (i.e., 0.40 and 0.25 µm), and we also provide preliminary data regarding aging tests performed under continuous illumination in ambient conditions without encapsulation. Appendix A shows the current–voltage characteristics of ZnO NW array-based perovskite solar cells for ZnO—0.25 µm and ZnO—0.40 **µm**, with the two different HTLs. The PV parameter results are presented in Table 8.

It is clear that, while increasing the length of the NWs from 0.25 µm to 0.40 µm, the device performance significantly increased from 2.7% to 4.9% (using spiro-OMeTAD-HTL), and 1.1% to 2.2% (using Cz–Pyr-HTL), respectively. This trend confirms that there is a compromise for NW length to enable efficient charge extraction before series or shunt resistances finally reduce the device performance (see Section 3.2). Regarding the nature of the HTL, Cz–Pyr-based devices exhibit much lower J_sc_ values compared to devices based on spiro-OMeTAD. Such observation can be related to a larger hole mobility for spiro-OMeTAD (2.5 × 10^−5^ cm^2^ V^−1^s^−1^) as measured using field-effect transistors in saturation regime, compared to Cz–Pyr (7.4 × 10^−6^ cm^2^ V^−1^s^−1^) [50]. The incident photon-to-current conversion efficiency (IPCE) measurements show that the use of 0.40 µm long NWs instead of 0.25 µm enhances the photocurrent in the whole spectral region under study (300–800 nm), with both spiro-OMeTAD and Cz–Pyr (see Appendix A). Such behavior is clearly consistent with a better hole mobility of spiro-OMeTAD compared to Cz–Pyr, and points out the positive influence for NW of moderate length. 

We finally conducted a preliminary aging test of the devices through continuous operation under solar illumination and without encapsulation for 20 min (Appendix A). Ambient conditions were used in this case, without thermal regulation at 25 °C, leading to a mean temperature of 55 °C and a moisture level of around 50% (conditions not finely controlled as in more adapted ISOS protocols) [79]. The Cz–Pyr-based device shows an interesting behavior compared to spiro-OMeTAD in these harsh conditions, with increasing performance mainly governed by J_sc_, unlike the spiro-OMeTAD device, which sees its performance slowly decreasing with time. While many mechanisms have been discussed [80,81] to interpret the evolution of the perovskite active layer in these conditions, explaining common losses in V_oc_ and FF [82], this strong difference in device behavior suggests a specific role played by the carbazole HTL in device degradation. While Cz–Pyr shows a relatively lower glass transition temperature than spiro-OMeTAD [50], these results indicate a beneficial interfacial configuration with the perovskite active layer, which positively affects the performance evolution with time. The role of the dopants (*tert*-butyl pyridine and lithium salt), which are known to be a major source of performance drop in the case of spiro-OMeTAD [75], remains to be clarified in this case as well.

The comparison of published AZO/ZnO NWs/perovskite-based device PV performances with our proposed device results is demonstrated in Table 9. In our study, we have not annealed the AZO substrate before making ZnO NWs growth which might be another possible reason for the lower PV performance than the published report, especially with V. L. Ferrara et al.’s results, because their observation shows that the annealed AZO substrate at 150 °C evidently enhances the crystal quality compared to non-annealed sample that helps to achieve a good crystal nucleation sites [74]. Also, the present contribution provides novel device architecture (i.e., AZO/ZnO NWs/perovskite/Cz–Pyr/Au) with a significant efficiency of 2.2%, half of conventional spiro-OMeTAD-HTL-based devices (see Table 9).

**Table 9 nanomaterials-12-02093-t009:** Published PV performance of AZO/ZnO NWs/perovskite architecture devices.

Device Structure	Jsc (mA cm^−2^)	Voc (V)	FF	PCE (%)	Ref
AZO/ZnO NRs/MAPbI_3_/Spiro-OMeTAD/Au	16.00	0.80	0.53	7.00	[74]
AZO/ZnO NRs/MAPbI_x_Cl_3-x_/Cu	14.87	0.86	0.28	3.62	[83]
AZO/ZnO NRs based DSSC	05.01	0.60	0.43	1.31	[84]
AZO/ZnO NWs/FACsPb(IBr)_3_/Spiro-OMeTAD/Au	16.10	0.64	0.47	4.9	This work
AZO/ZnO NWs/FACsPb(IBr)_3_/Cz–Pyr/Au	07.00	0.81	0.39	2.2	

## 4. Conclusions

This study focused on the ZnO NWs growth over the AZO substrate using the low-temperature hydrothermal growth (HTG) method to fabricate efficient perovskite solar cells based on AZO/ZnO NWs/perovskite/HTL (spiro-OMeTAD or Cz–Pyr)/Au device configuration. The effect of several parameters, especially HTG precursors concentrations such as zinc nitrate hexahydrate, HMTA, PEI, and growing time, were systematically studied to achieve dense and well-aligned ZnO NWs array with characterized void fraction in order to boost the PV device performance. Results highlight that a longer growing time (i.e., 15 h) and higher zinc salt concentrations (i.e., 35 mM and 50 mM) are unfavorable to the device performance. Also, AZO/ZnO NW-based devices demonstrate superior performance than AZO/ZnO or AZO/SnO_2_ compact ETL-based devices. Finally, Cz–Pyr-HTL-based AZO/ZnO NWs device provides comparable performance to the conventional spiro-OMeTAD-HTL devices. Overall, the obtained solar cell efficiencies are lower than the published conventional perovskite device configurations, therefore, enhancing the ZnO NWs layer surface homogeneity, especially the NWs’ alignment with an appropriate gap, will be a beneficial strategy to further increase the device efficiency in the near future. This current research observations suggest that using ZnO NW over AZO substrate is an effective approach for the perovskite solar cells in terms of performance and stability. 

## Data Availability

The data presented in this study are available from the corresponding author upon request.

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
