# Peer review of "Low-Temperature Hydrothermal Growth of ZnO Nanowires on AZO Substrates for FACsPb(IBr)3 Perovskite Solar Cells"

_nanomaterials, 2022, doi:10.3390/nano12122093_

Round 1

Reviewer 1 Report

For the nanowire synthesis, the variations in both morphology and dimension (like diameter) are quite large. Please comment on this.

Following the above question, what is the I-V characteristics among different cell devices? Please also comment on the performance variations.

ZnO NW growth using hydrothermal method is quite mature, and there lacks sufficient introduction of the intrinsic advantage of the experimental approaches used here. Also, a performance comparison with other reported devices is also suggested for better demonstration of the advantages and novelty in this work.

The format of the equations in the manuscript should be corrected.

Reviewer 2 Report

This report makes a contribution to the development of perovskite solar cells. A ZnO nanowire structure is suggested (with low T hydrothermal synthesis) and demonstrated in practice. The nanowire properties are tuned to improve cell performance. ZnO grow processes are discussed. This work is original and publishable.

The manuscript is well written and structured. The supporting information are useful. Good quality figures and illustrations support the text. Literature citations are appropriate and sufficient. Experimental information is provided. There are only minor suggestions for revision:

(A) the term perovskite solar cell is not specific enough. Title and abstract do not state which perovskite (FAI?) and therefore some modification is requested (for both title and abstract) to be more clear about which type of perovskite was used.

(B) there is a problem with scheme 1, probably due to the pdf conversion.

(C) the efficiencies reported here are clearly at the lower end. Why is this? Maybe some general text can be added in the conclusions?

Round 2

Reviewer 1 Report

The previous concerns have been properly addressed. It can be accepted for publication in the present form.